# Meteorological, soil moisture, surface water, and groundwater data from the St. Denis National Wildlife Area, Saskatchewan, Canada

Edward K.P. Bam[1], Rosa Brannen[1], Sujata Budhathoki[1], Andrew M. Ireson[1], Chris Spence[2], Garth van der Kamp[1,2]

[1]Global Institute for Water Security, University of Saskatchewan, Saskatoon, S7N3H5, Canada

[2]National Hydrology Research Centre, Environment and Climate Change Canada, Saskatoon, S7N3H5, Canada

Correspondence to: Edward Bam (edward.bam@usask.ca) and Andrew Ireson (andrew.ireson@usask.ca)

**Abstract.** Long-term meteorological, soil moisture, surface water and groundwater data provide information on past climate change, most notably information that can be used to analyze past changes in precipitation and groundwater availability in a region. These data are also valuable to test, calibrate and validate hydrological and climate models. CCRN (Changing Cold Regions Network) is a collaborative research network that brought together a team of over 40 experts from 8 universities and 4 federal government agencies in Canada for 5 years (2013-18) through the Climate Change and Atmospheric Research (CCAR) initiative of the Natural Sciences and Engineering Research Council of Canada (NSERC). The working group aimed to integrate existing and new data with improved predictive and observational tools to understand, diagnose and predict interactions amongst the cryospheric, ecological, hydrological and climatic components of the changing Earth system at multiple scales, with a geographic focus on the rapidly changing cold interior of Western Canada. The St. Denis National Wildlife Area database includes atmosphere, soil, surface water and groundwater data for the prairie research site. The meteorological measurements are observed every 5 seconds, and half-hourly mean values (or total values) are logged. Soil moisture data comprises volumetric water content, soil temperature, electrical conductivity and matric potential for probes installed at depths of 5 cm, 20 cm, 50 cm, 100 cm, 200 cm and 300 cm in all soil profiles. Additional data on snow surveys, pond and groundwater levels, surface water chemistry and water isotopes collected on an intermittent basis between 1968 and 2018 are also presented, including information on the dates, ground surface elevations (in meters above sea level), and geographical positions coordinates used to construct hydraulic heads. The spatial data tables provide the geographical coordinates of piezometers and wetlands that are relevant to the interpretation of the records and the digital elevation map (DEM) of the site and surrounding landscape. The compiled data are available at https://doi.org/10.20383/101.0115.

## 1 Introduction

The Canadian portion of the North American Prairies is bounded to the west by the Rocky Mountains, and the north and east by the Boreal Forest. It contains around 80% of Canada's agricultural area, with an annual crop market value in 2016 of $41 billion (Statistics Canada, 2017). The region is characterized by a semi-arid, seasonally frozen climate, with colder and wetter conditions moving from southwest to northeast. The surficial geology is dominated by glacial till, interspersed with glacio-lacustrine sand and gravel deposits, and the glaciated landscape is generally flat or gently rolling, with thousands of depressions. Due to the low amounts of precipitation, drainage networks are poorly developed, and precipitation excess tends to form lakes and wetlands in surface depressions (Hayashi et al., 2016). Surface water bodies in the depressions have widely varying sizes, from large permanent lakes, such as Redberry Lake, Saskatchewan (van der Kamp et al., 2008), to ponds that rarely dry out, to ephemeral ponds that annually dry out annually. Farming practices over the past century include widespread artificial wetland drainage in some portions of the region (Rashford et al., 2011).

The St. Denis National Research Area (Figure 1) was established in 1968 by the Canadian Wildlife Service for research on how agricultural practices affect waterfowl production. The site was primarily selected because it was a cultivated land parcel with an abundance of wetlands close to the City of Saskatoon. The St. Denis National Research Area was officially listed in the Canada Gazette with an amendment to the Wildlife Area Regulations in 1978 (SOR/78-466). The St. Denis National Research Area is the only site in the Environment and Climate Change Canada protected areas network to bear the title of "National Research Area", however, the colloquial name of the reserve is St. Denis National Wildlife Area (SDNWA).

At the time of establishment, nearly 60% of the 361.5 hectares in the SDNWA were under cultivation, with the remainder a mix of native grasslands, shrublands, and wetlands. No wetland drainage has occurred on the site since 1968. Lease agreements with local farmers are used to keep the annual cropland in production with wheat, barley, and canola, while the remainder of the vegetation remains undisturbed. In 1977, a program was initiated to convert 97 hectares of cultivated land to a perennial forage mix of smooth brome (*Bromus inermis*) and alfalfa (*Medicago sativa*) (van der Kamp et al.,1999). The program was to minimize erosion on sloping and light textured soils from tillage activities, to provide nesting cover and food for wildlife, and was consistent with land use changes in the surrounding region. There were minor changes in land cover until 2015 when a seeding program began to convert the remaining cropland back to permanent grassland cover.

Much inventory, monitoring, and research have occurred at SDNWA, consistent with the original purpose of the site. Research efforts have been diverse and growing, with an emphasis on migratory bird ecology, wetland hydrology, and soil science. SDNWA has grown in value to the research community because of the accumulated long-term data collected consistently. It is a site that is unrivaled in the Canadian Prairies for the length and breadth of hydrological observations. Pennock et al. (2013) provide a good summary of the site and the research history.

The SDNWA and surrounding landscape area is hummocky, with local relief of the order of 15m. The area lies within and near the lower end of a 2,400 ha closed watershed for which detailed Lidar elevation data are available. Aspen bluffs occupy some of the larger depressions in the drainage basin. Soils are predominantly Dark Brown Chernozems and Orthic Regosols; both developed from glacial till (Miller et al., 1985). The glacial stratigraphy of the site is well-documented: roughly one hundred meters of clay-rich glacial till of low permeability lies under the soils, interspersed with isolated sand lenses and a continuous thin layer of sand at about 25 to 30 m depth (Hayashi et al., 1998).

Over the past 50 years, field research has been undertaken at SD NWA by hydrologists, hydrogeologists, biologists, ecologists, soil scientists and others, and there have been more than 100 papers published on this site. Many of these studies were based on short-term field experiments, which have produced snapshots of data in time. Here, we focus on the continuous monitoring program, which is ongoing and is the foundation of much of the short-term research. The longest records are of pond levels and chemistry, dating back to 1968. Soil and groundwater observations have been recorded continuously since 2013 (before this there are large amounts of groundwater data, but those observations are not continuous to present).

**Figure 1: Map and location of St. Denis National Wildlife Area. The grey area on the inset map represents the extent of Prairie Pothole Region in North America.**

## 2 Meteorological data

There are two climate stations at the St. Denis National Wildlife Area. The older, a 10 m mast tower, is at an elevation of 554 masl at 52.2037°N 106.1067°W located on the upland south of Pond 50 surrounded by brome-alfalfa grassland (Figure 1). This station was initially deployed in the spring of 1989 as part of the Wetlands Ecosystem Vulnerability Study (WEVS). It was initially used to evaluate the water budgets of prairie wetlands, particularly Pond 50 (Woo and Rowsell, 1993). From 1991 to 1998 operation of the station was transferred to the Meteorological Service of Canada. In 1998, the equipment was transferred to the Canadian Wildlife Service (CWS), for continuing use as a WEVS facility. Reorganization of the CWS in 2006 prompted responsibility of the tower to be transferred to the Water Directorate of the Science and Technology Branch of Environment and Climate Change Canada. These transfers sometimes resulted in substantial breaks in the period of record. In 2011, a second climate tower was constructed that was capable of supporting eddy covariance equipment for measuring turbulent energy and carbon fluxes and all component net solar radiation. This is a 10 m scaffolding tower located at 52.2089°N 106.0889°W at a local high point in a field currently supporting dense nesting cover (Figure 1). The specifics of each tower, including variables and units and sensor types and heights, are summarized in Table 1. Three-dimensional wind speeds, air temperature, water vapour content, and carbon dioxide density are measured at 10 Hz with turbulent fluxes calculated over a half hour period on the Campbell Scientific CR3000 logger. The turbulent flux data within these records is uncorrected and

should be treated with caution. All meteorological data are observed every 5 seconds, and half-hourly averages (or totals) are currently logged on Campbell Scientific CR1000 (mast tower) and CR3000 (scaffolding tower) data loggers. Rainfall data, measured with Texas Electronics TE525M tipping bucket rain gauges, are reported as the hourly or half-hourly totals. It is important to note that while the tipping bucket rain gauges are in operation all year and may record precipitation during the winter months, they only supply reliable measurements of liquid precipitation when not accompanied by freezing or frozen precipitation. The Figure 2 shows total and accumulated precipitation, evaporation and the temperature records at 2 m for the at the SDNWA.

Figure 2: Meteorological data from SDNWA: a) the total annual rainfall for the period of record (1991-2017) and annual snowfall (only available 2014-2017); b) cumulative total precipitation (rain and snow) and evaporation for the 2016-2017 hydrologic year; c) mean monthly air temperature at 2 m for the period of record (1991-2017); and d) daily air temperature for the 2016-2017 hydrologic year.

## 3 Snow surveys

Snow surveys have been conducted at SDNWA by Environment and Climate Change Canada (ECCC) since 1994. Snow depths and densities are measured along three transects (Figure 1) and converted to snow water equivalent (SWE) and then generalized over the whole site (Figure 3). The snow surveys are performed once each year in March to estimate the SWE on the ground before snowmelt begins. One of the snow surveys consists of two perpendicular transects that cross in the centre of Pond 109. Each transect is approximately 250 m. Snow depth is measured at every 5 to 10 m. At every fifth depth measurement, a density sample is collected using an Eastern Snow Conference snow tube. This survey measures snow accumulation in the grass, wetland vegetation, trees, and an ice-covered pond. The other snow survey is a 700 m transect that runs east to west through cropped and grass fields across the western portion of the SDNWA. Snow depths are measured every 15 - 20 m and density samples are taken every fifth depth measurement. SWE values are calculated for each point and then averaged to calculate mean SWE for each land cover. The average SWE over the entire SDNWA is estimated by calculating a weighted mean by a land cover fraction of the total area.

**Figure 3. Weighted mean of snow water equivalent (SWE) of snowpack before melt over SDNWA. The blue line represents the 24-year mean of snow accumulation.**

## 4 Pond water level, chemistry and bathymetry data

At the St. Denis National Wildlife Area pond water depths in wetlands have been monitored since 1968. These wetlands generally have areas between 0.1 and 10 hectares and contain ponds that range in class from vernal to ephemeral to permanent. SDNWA wetlands were initially mapped based on aerial photography (Hogan and Conly, 2002) and numbered. The major

wetlands were numbered 1 to 147 and smaller wetlands adjacent or adjoined to the larger depressions were sub-labeled with letters a, b, c, and d.

The record of pond depth measurements dates from 1968 to 2017 and includes measurements from 135 ponds. Canadian Wildlife Service staff began monitoring 76 of the SDNWA wetlands in 1968; an additional 57 started being monitoring after 2011. The frequency of measurement varies annually and among wetlands. Monitoring typically begins in spring and extends through to the fall. Measurements are not conducted during the winter because many ponds will freeze to the sediment and because snow and ice accumulation on ponds influence water depth under ice. There are some wetlands with data available for every year from 1968 to the present (Pond 25, Pond 50, Pond 65, Pond 90, Pond 109, and Pond 120). Many were dry during the drought of 1999-2002. In 2015, a long-term monitoring plan was developed jointly with researchers from the University of Saskatchewan and ECCC to monitor pond depths, stable isotopes and water quality of 25 wetlands once each in May, July, and October.

The methodology for collecting pond depth data is described in Conly et al. (2004). The lowest bottom elevation in the wetland depression is used as the relative datum and the geodetic elevations of these points have been determined for many of the wetlands. Measurements are made by wading into the pond and using a measuring rod to measure water depth at monitoring markers (usually a metal T-bar installed deep into pond sediments to prevent heaving or movement). The measuring rod is attached to a 6-cm diameter circular base to prevent the rod from being pushed into the sediment. Shallow seasonal and ephemeral ponds require only a single marker. Deeper ponds that vary considerably in flooded area and depth have multiple markers installed at various elevations to ensure a measurement can be made when markers installed at lower elevations are flooded. Depth measurements are taken at the same time at multiple markers to ensure markers are tied to the local datum. Point measurements at single markers (in smaller wetlands) are generally within 25 mm of those measured with conventional survey equipment and benchmarks. The accuracy in larger wetlands is considered to be within 50 mm (Conly et al., 2004). The year-to-year and seasonal variations in pond hydrographs are apparent from the long-term record (Figure 4).

Figure 4: Hydrographs of ponds in Wetlands 1, 50 and 109.

Water chemistry of ponds and groundwater have been measured in a number of studies at St Denis National Wildlife Area (e.g., Hayashi et al., 1998; Berthold et al., 2004; Waiser 2006; Heagle et al., 2007, 2013; Pennock et al., 2010, 2013). The record consists of measurements of individual ions and overall solute load measures, i.e., electrical conductivity (*EC*, Figure 4) and total dissolved solids (*TDS*). The details of the equipment and methods are described in Hayashi et al. (1998), Berthold et al. (2004), Waiser (2006), Heagle et al. (2007& 2013) and Pennock et al. (2010).

Figure 5: Time series of pond water electrical conductivity (log scale) at St. Denis showing multi-year variations in pond water chemistry.

Detailed elevation surveys were carried out in 1994, 1998, 1999, 2008, and 2009 on twenty-six wetlands depressions using electronic total stations within the St. Denis NWA. The catchment extent of the wetlands and depressions, vegetation, soil, and the hydroperiods of ponds in the respective wetlands are published in Hayashi and van der Kamp (2000) and Minke et al. (2010). Survey points were spaced horizontally at 10–15 m intervals in the uplands, 5–10 m intervals in the wetlands, and 2–5 m intervals in small depressions. The estimated measurement error is within a few centimeters for elevation and within a few tens of centimeters for horizontal location (Hayashi and van der Kamp, 2000; Minke et al., 2010). The water depth data for these wetlands (Figure 4) can be presented as water surface elevation for comparison with groundwater levels.

## 5 Soil moisture data

There are four soil moisture profiles at SDNWA where continuous observations of water content, matric potential, temperature, and electrical conductivity have been made since July 2013. Three of these profiles are located along a lowland transect between ephemeral ponds (107 and 108a) adjacent to Pond 109 and one profile is located towards the top of a hillslope east of Pond 1 (Figure 1). In each case, piezometers are collocated with the soil moisture profiles, as described below.

Hydra Probe sensors (Hydra II, Stevens Water Monitoring Systems Inc, 2007) are impedance sensors that simultaneously measure soil volumetric water content, soil temperature, and electrical conductivity. The probes were installed at depths of 5 cm, 20 cm, 50 cm, 100 cm, 200 cm and 300 cm in all soil profiles. During soil freezing the dielectric constant of ice is much lower than that of liquid water, so the instrument is likely to give a reasonable measure of the liquid water content (Spaans and Baker, 1995). The soil matric potential is measured using the heat dissipation sensors (229 probes, Campbell Scientific) at depths of 5 cm, 20 cm, 50 cm, and 100 cm. The heat dissipation sensor is a porous block sensor which measures the pressure of dry soil and has a working range of -10 kPa to -2500 kPa. Tensiometers (T4e, Decagon Devices) are used to measure the pressure of wet soil at the depths of 100 cm, 200 cm, and 300 cm. The working range of the tensiometer is +100 kPa to -85 kPa. The soil freezing temperature and moisture content for three profiles at 20 cm near Pond 109 is shown in Figure 5(a-b). Figure 7 shows the observed soil water content and soil freezing characteristic of Transect 1 (Figure 1) at 5 cm and 20 cm depths at St. Denis.

**Figure 6: Soil freezing depth for three profiles near Pond 109.**

**Figure 7: Observed soil water content and soil freezing characteristic of Transect 1 at 5 cm and 20 cm depths at St. Denis.**

## 6 Groundwater level data

The hydrogeology of St. Denis is reasonably complex, comprising a shallow and relatively high permeability weathered till layers overlying unweathered till aquitards and coarse-grained inter-till confined aquifers. A large number of piezometers were installed in all three of these units, mostly for individual short-term projects, and as a result most of the data are discontinuous. These data are none-the-less available and provide useful insights into the spatially and temporally variable interactions between groundwater and surface water bodies. Continuous monitoring of the water table started in 2013 at a piezometer near Pond 109 collocated with the soil moisture transects. Two piezometers were installed in a confined aquifer and instrumented in November 2013. These are the deepest piezometers on the site (39 and 41 m below ground level). Unvented Solinst Levelloggers are used to monitor water levels, corrected for changes in barometric pressure with a Solinst BaroLogger placed at the scaffolding climate tower. The Figure 8 shows ground and surface water levels at SDNWA for 2014 to 2016.

**Figure 8: Groundwater and surface water levels in St. Denis.**

## 7 Water isotope data

Stable isotopes of water ($\delta^{18}O$ and $\delta^2H$) have been measured sporadically for some of the ponds at St. Denis since 1993. These data are complemented by campaign-based samples of rainfall, snow (mainly in the period 2013-2014) and groundwater (sporadic measurements between 1993 and 2014, and covering depths between 1.2 – 41 m below ground level). Furthermore, continuous snow and rainfall samples from Saskatoon, 35 km away, are available for the period 1993 to 2014. Isotope ratios of $\delta^2H$ and $\delta^{18}O$ of all samples were measured at the National Hydrology Research Centre in Saskatoon, Canada. The sampling and analysis methodology is described in detail in Lis et al. (2008). Isotope ratios of tritium were measured at the Rafter Radiocarbon Laboratory, National Isotope Centre, GNS Science in New Zealand. The stable isotope ratios of water were analyzed using three instruments: Micromass Optima continuous flow mass spectrometer, Micromass IsoPrime dual inlet/continuous flow mass spectrometers, and off-axis integrated cavity output spectroscopy (OA-ICOS) laser.

Between 1993 and 2007, the precipitation, ponds, and groundwater samples were analyzed for $\delta^2H$ and $\delta^{18}O$ using the isotope-ratio mass spectrometers (IRMS). The protocols for the isotope-ratio mass spectrometer analysis follow the standard methods (Begley and Scrimgeour, 1997; Coleman et al., 1982; Eiler and Kitchen, 2001; Epstein and Mayeda, 1953; Karhu, 1997; Kelly et al., 2001; Socki, 1999). The $^{18}O/^{16}O$ ratio in the water samples were analyzed by the equilibration of water samples with $CO_2$ gas at $25 \pm 0.1$ °C for 24 h to produce $CO_2$ gas (Epstein and Mayeda, 1953). After the equilibration, the $CO_2$ gas was extracted and purified through a vacuum cryogenic line. The $^2H/^1H$ ratio was analyzed by the production of hydrogen gas using Cr at 850 °C (Coleman et al., 1982). The $^{18}O/^{16}O$ and $^2H/^1H$ ratios were measured against the internal standards which

were calibrated using Vienna-Standard Mean Ocean Water (V-SMOW) and Standard Light Antarctic Precipitation (SLAP). The results were normalized to the VSMOW-SLAP scale and reported in the delta notation as decribed in Coplen (1988). The analytical reproducibility is ± 0.1 ‰ and ±1.0 ‰ for $\delta^{18}O$ and $\delta^{2}H$. Both offline dual-inlet (i.e., zinc reduction or uranium reduction) and continuous flow IRMS methods (i.e., Cr reduction or C) have measurement accuracies of the order of (± 0.5 to

± 4 ‰) for $\delta^{2}H$ and the automated CF-IRMS methods, such as C-reduction to CO and $CO_2$-$H_2O$ equilibration, are between ± 0.1 and ± 0.4 ‰ for $\delta^{18}O$ (Lis et al., 2008).

A Los Gatos Research DLT-100 liquid water isotope analyzer system coupled with a CTC LC-PAL liquid autosampler (Los Gatos Inc., California) was employed in stable isotope analyses done between 2008 and 2018. The analysis follows the methods

described in Lis et al. (2008) and IAEA manual (2009). Laboratory standards (INV1: $\delta^{2}H$ = -220.0 ‰, $\delta^{18}O$ = -28.5 ‰ and ROD3: $\delta^{2}H$ = -8.0 ‰, $\delta^{18}O$ = -1.2 ‰) were calibrated with Vienna-Standard Mean Ocean Water (V-SMOW2) and Standard Light Antarctic Precipitation (SLAP2) reference waters.  INV1 and ROD3 were used to normalize the results to the VSMOW2-SLAP2 scale by assigning $\delta^{18}O$ and $\delta^{2}H$ values of -55.5 ‰ and -428.0 ‰, respectively, to the SLAP2 reference water. Consequently, all measured values reported are relative to VSMOW/SLAP $\delta$ scale. Samples, standards and control samples

(river water) were analyzed repeatedly six times. The laboratory precision was ±1.0 ‰ for $\delta^{2}H$ and ± 0.2 ‰ for $\delta^{18}O$.

Tritium samples ($^{3}H$) samples were shipped to Rafter Radiocarbon Laboratory, National Isotope Centre, GNS Science in New Zealand for ultra-low-level tritium counting using electrolytic enrichment and liquid scintillation counting (LSC). The $^{3}H$ concentrations are expressed in tritium units (TU); the precision at an average tritium concentration of New Zealand rain of 4

TU is ± 0.06 TU (98.5 %), and the detection limit is ±0.025 TU (Morgenstern and Taylor, 2009). Details of the analytical procedure are provided in Morgenstern and Taylor (2009). A summary of the stable isotope data is shown in Figure 9.

**Figure 8: A scatter plot of $\delta^{2}H$ versus $\delta^{18}O$ at St. Denis, Saskatchewan.**

**8 Spatial data**

Elevation data of the SDNWA and surrounding area were collected during a Light Detection and Ranging (Lidar) survey conducted by the Canadian Consortium for Lidar Environmental Applications Research (C-CLEAR) on 9 August 2005. The exact methods of the Lidar data survey and processing of the digital elevation model (DEM) are described in Toyra et al. (2008). The generated DEM is provided in UTM Zone 13 NAD83 and the elevations are orthometric heights based on the

CGG05 geoid model. The elevations are tied to a local benchmark near Pond 1. The ground data points were interpolated into a 1 m DEM using the Inverse Distance Weighted (IDW) algorithm. The Lidar DEM was evaluated based on in situ survey

data (GPS points and total station surveys). The results indicate that the accuracy of LiDAR DEM in the agricultural fields or grasses is 0.13m, while the accuracy in the shrubs and trees surrounding the wetland ponds is 0.17m.

## 9 Overview of content

The record from both climate towers shows strong seasonality of incoming radiation and turbulent fluxes at the SDNWA. Maximum incoming solar radiation (typically ~ 350 Wm$^{-2}$) occurs in June near the summer solstice. Conversely, minimum solar radiation of 10 W m$^{-2}$ is in December. There is very little seasonal lag between incoming solar radiation and net radiation, which ranges from 200 W m$^{-2}$ to -30 Wm$^{-2}$ between June and December. The record does include several spikes in the radiation and turbulent flux data that should be removed prior to any analysis. The mean of hourly wind speed over the period of record (1992 - 2018) is 3.8 ms$^{-1}$. The period of record includes half-hourly mean wind speeds as high as 16.9 ms$^{-1}$. Daily maximum air temperatures often reach +27 °C in July and can be well below -30 °C anytime between November and February (Figure 9). These cold air temperatures are also associated with dry air with relative humidity often less than 50% in late winter. The measurements of relative humidity were some of those that suffered from the changing agencies responsible for the towers. The near 30-year length of the record provides an excellent perspective on the cycles of rainfall in the central Canadian Prairie. The record drought from 1999-2002 (van der Kamp and Hayashi, 2009) during which annual rainfall averaged 20 mm is well documented and is in stark contrast to a recent wet period (Dumanski et al., 2015) when annual rainfall averaged 330 mm from 2005-2013. A recent addition to the towers was a Geonor total precipitation gauge in 2015, but the record is too short to allow for a description of the entire precipitation regime at the site.

The mean maximum spring SWE as measured during the March snow surveys was 62 mm throughout the recorded period (1994-2017). There was a sustained period of lower than average SWE from 1998 to 2003, followed by an above-average period from 2004-2009. The spring snowpack in 2013 was 120 mm and the largest in the 24-year period.

The hydrographs of wetlands 1, 50 and 109 follow a distinct annual cycle that has a peak in pond level at the end of the snowmelt period followed by a gradual decline during summer months. Wetland 109 contains a seasonal pond that dries out completely in some summers; may carry water over between years during wetter periods and has reached spill elevation occasionally in recent wet years. The larger wetlands, 1 and 50, have permanent and semi-permanent ponds, respectively. Wetland 1 receives inflow in most years from a roughly 1000 ha watershed and spills to lower-lying ponds in most years, while Wetland 50 has never spilled during the period of record.

Wetland pond chemistry in the prairies is controlled primarily by water balance components of evaporation, plant transpiration and groundwater discharge. The long-term fluctuations in the chemistry of wetland ponds are caused by multi-year, wet-dry cycles associated with meteorological forcing and land-use in the region (Cressey et al., 2016; Goldhaber et al., 2016, 2014;

LaBaugh et al., 2016, Nachshon et al., 2014). During extreme wet conditions groundwater discharge and surface runoff of salts from an upland area into freshwater ponds results in freshwater ponds becoming salinized (e.g., Pond 109 and 120, Figure 4). In addition to these water balance controls, geochemical and biochemical interactions within a wetland pond and underlying wetland soils also add or remove solutes from the wetland pond water (Heagle et al., 2007; LaBaugh et al., 2016; Pennock et al., 2014). Inadequate knowledge of the climate history and land use change around a particular pond could lead to a misinterpretation of the pond's hydrological function by short-term observations of pond chemistry.

The maximum frost table depth is ~1.2 m and corresponds with cold surface air temperature and low snowpack accumulation (Hayashi et al., 2003). The frost table depth is affected by antecedent soil water content during fall before the freeze (Pan et al., 2017). The saturated water content at freeze-up is usually 0.5 with a high residual of liquid water content during frozen conditions. The variation in the water content values in different years might be attributed to heterogeneity and hydrological variation within the study area.

The stable water isotope compositions of precipitation, snowmelt, groundwater and surface water (ponds) are similar to isotope values of water taken from surface ponds and glacial deposits throughout southern Saskatchewan and central Canada (Fortin et al., 1991; Fritz et al., 1987; Jasechko et al., 2014, 2017; Kelley & Holmden, 2001). The data show distinct differences between the different sources. The precipitation data from SDNWA fall on the Saskatoon local meteoric water line (LMWL), but show distinct seasonal variability with winter measurements (snow) more depleted than summer rainfall. The pond water isotopes show evaporation like many surface waters. The shallow groundwater data are relatively similar to the pond data, but subject to less evaporation. The intertill aquifer data is biased towards the snow end of the spectrum of precipitation.

## 10 Data availability

The SDNWA dataset is stored at the Federated Research Data Repository (FRDR) and can be accessed from the FRDR at: https://doi.org/10.20383/101.0115.

## 11 Final remarks

The data from the SDNWA have contributed significantly to our understanding of groundwater-surface water interactions in prairie environments. The long-term dataset can be used to examine the inter-annual variability of hydrological fluxes, climate change impact on wetlands and groundwater resources. The unique dataset will be valuable to prairie hydrological research communities for various purposes such as inter-site comparison of hydrogeological processes or hydrological model testing.

## 12 Competing interests

The authors declare that they have no conflicting interest.

## 13 Acknowledgment

The field program at SDNWA was assisted by field assistants, scientists, and institutions who are too many to name. We especially grateful to the graduate students and post-doctoral fellows from the University of Saskatchewan who conducted hydrological research projects at the SDNWA over the years. We also thank research and field technicians who took responsibility for data collection and quality control, especially Randy Schmidt, and Branko Zdravkovic and Amber Peterson who assisted with data transfer and archiving. The program has been funded by Natural Sciences and Engineering Research Council (Discovery Grant, CCRN), National Hydrology Research Institute, Saskatoon, Global Institute for Water Security, University of Saskatchewan, and Environment and Climate Change Canada.

## 14 Author contributions

EKPB provided the stable water isotopes and groundwater data, described sample collection, measurement, summarized the entirety of all the data and put together the final drafts of the manuscript for comments. RB put together pond levels and snow surveys and meteorological data and plots, and aerial map for SDNWA. SB compiled the soil moisture data, created plots and wrote on instrumentation for the soil moisture data. AI discussed the relevant context of the hydrological data and wrote on the site description and hydrogeology, and provided edits and comments on the manuscript. CS wrote the introduction, described the instrumentation for the meteorological data and provided editorial comments on drafts. GV read through the final drafts of the manuscript and provided editorial help and reviews.

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

25  **Figures**

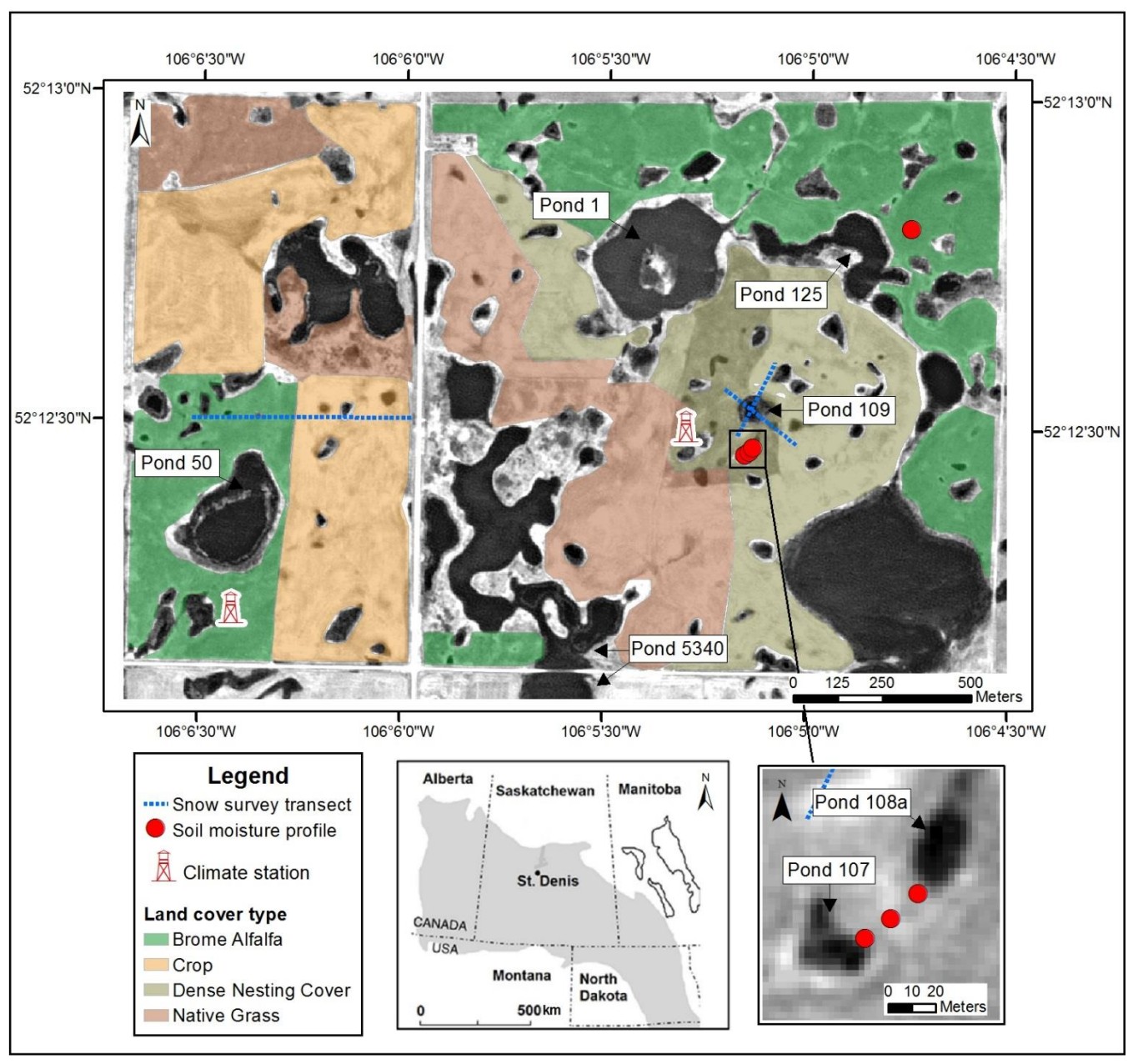

**Figure 1: Map and location of St. Denis National Wildlife Area. The grey area on the inset map represents the extent of Prairie Pothole Region in North America.**

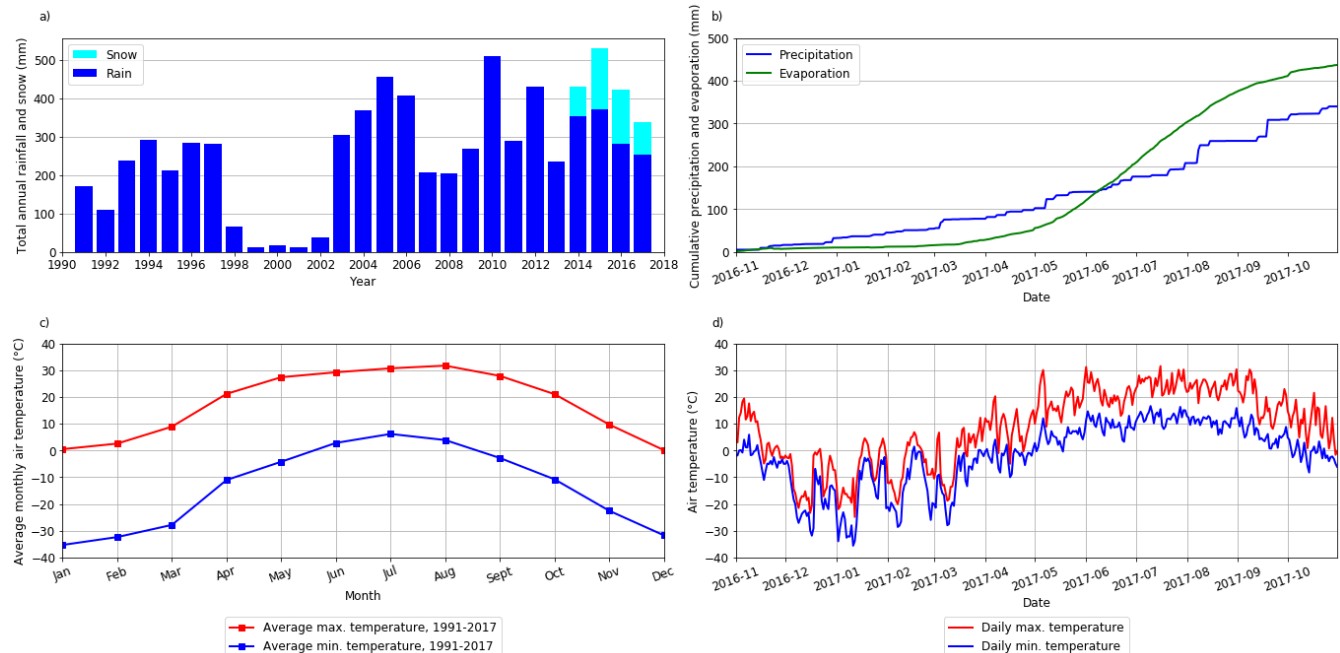

**Figure 2: Meteorological data from SDNWA: a) the total annual rainfall for the period of record (1991-2017) and annual snowfall (only available 2014-2017); b) cumulative total precipitation (rain and snow) and evaporation for the 2016-2017 hydrologic year; c) mean monthly air temperature at 2 m for the period of record (1991-2017); and d) daily air temperature for the 2016-2017 hydrologic year.**

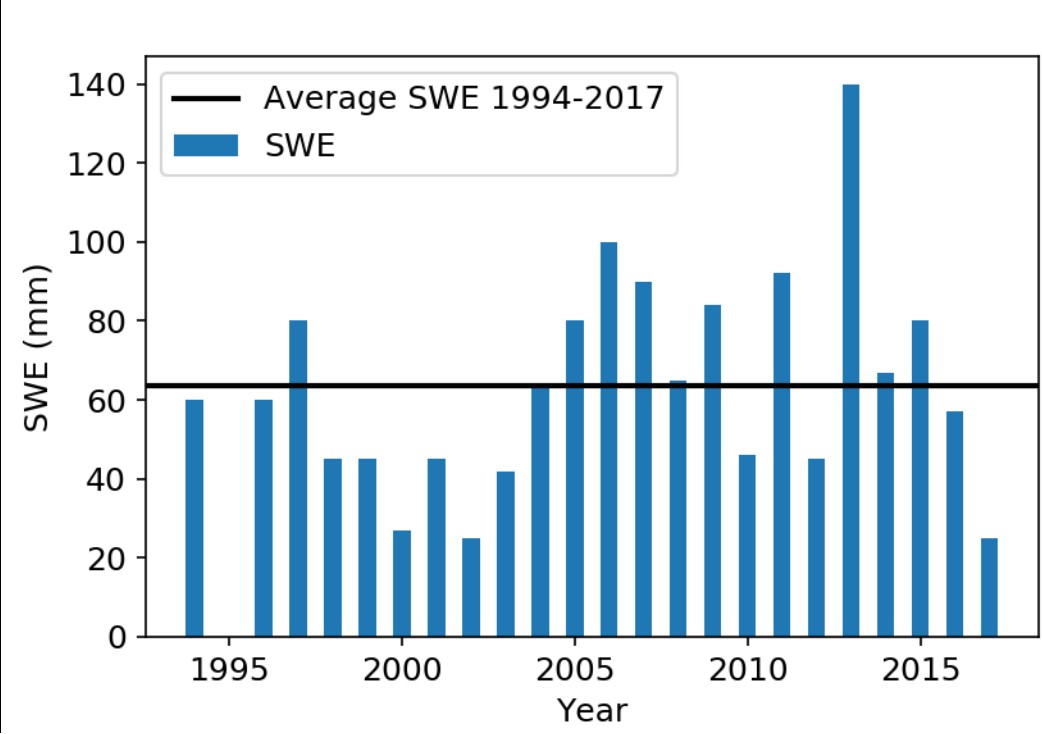

**Figure 3: Weighted mean snow water equivalent (SWE, blue bars) of snowpack before melt over SDNWA. The black line represents the 24-year average of snow accumulation.**

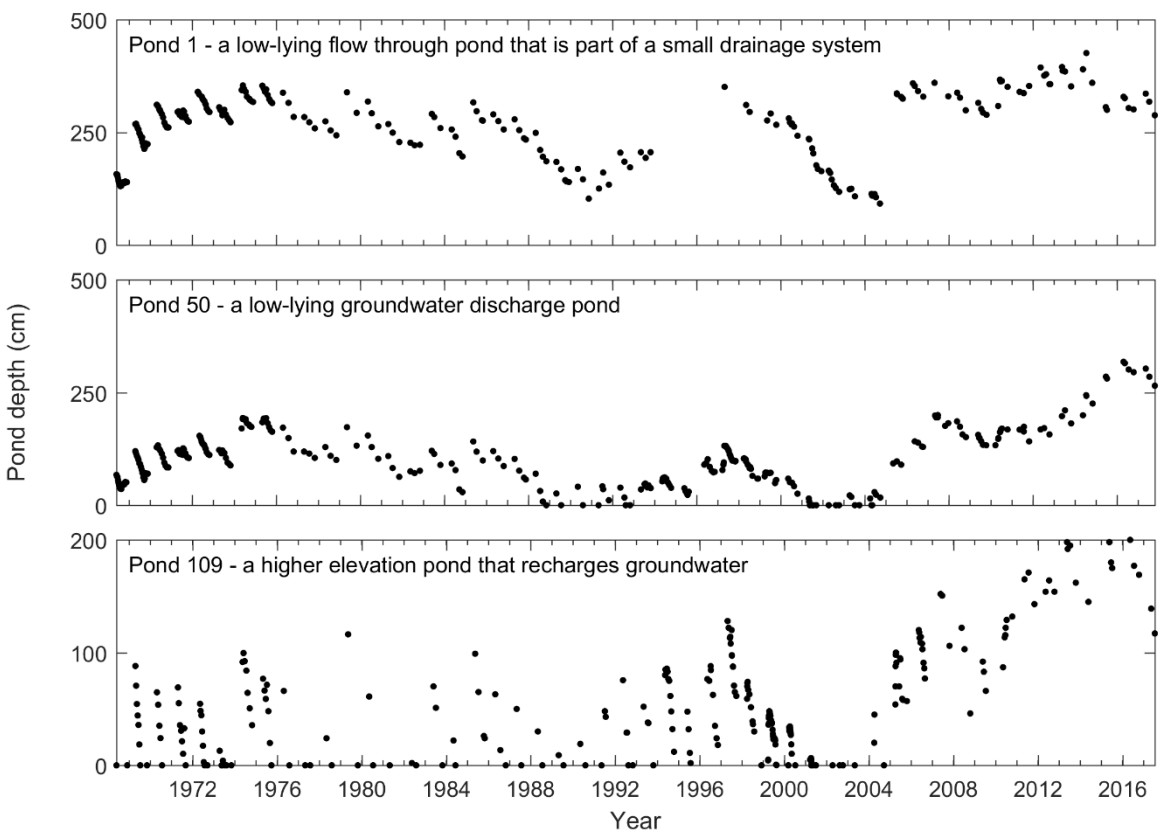

**Figure 4: Hydrographs of ponds in Wetlands 1, 50 and 109.**

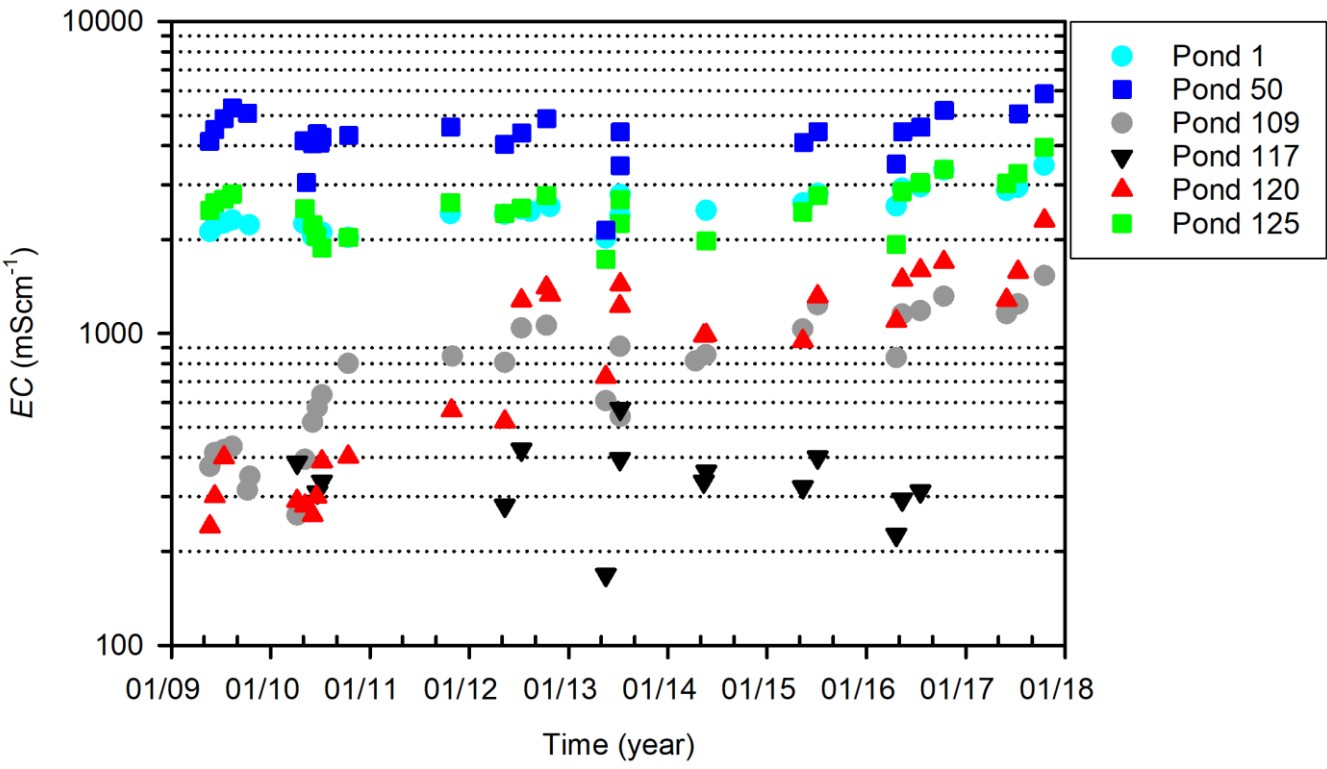

Figure 5: Time series of pond water electrical conductivity (log scale) at St. Denis showing the variations in pond salinity.

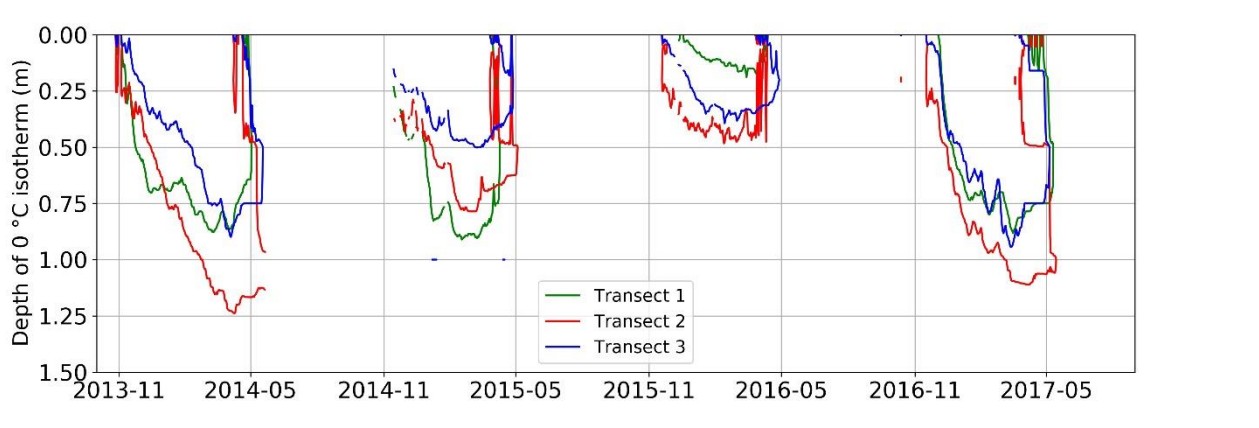

**Figure 6: Soil freezing depths in three soil profiles near Pond 109.**

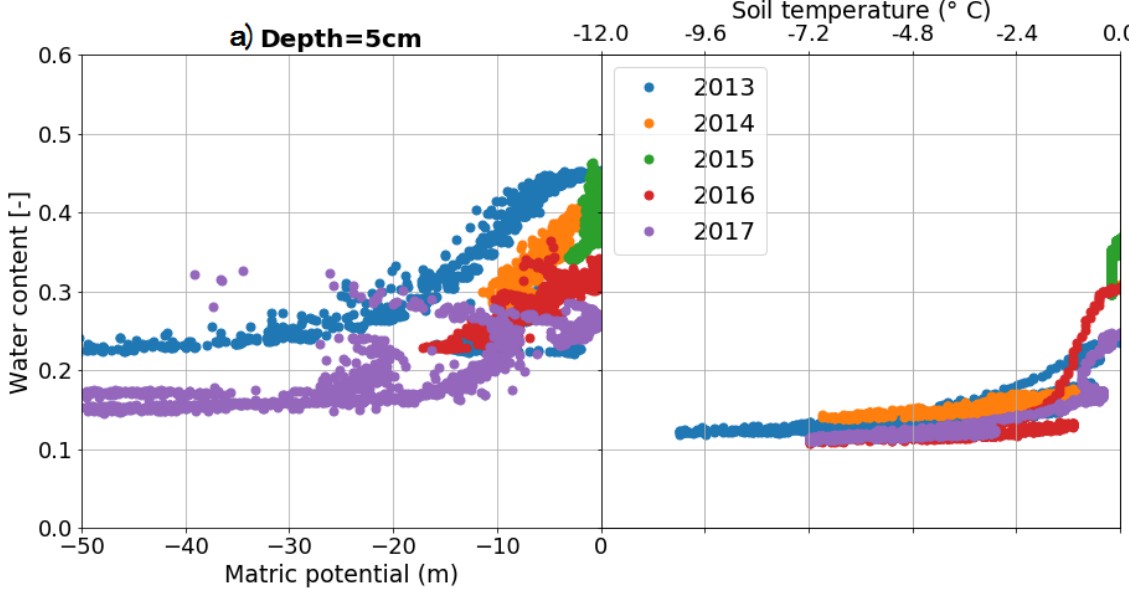

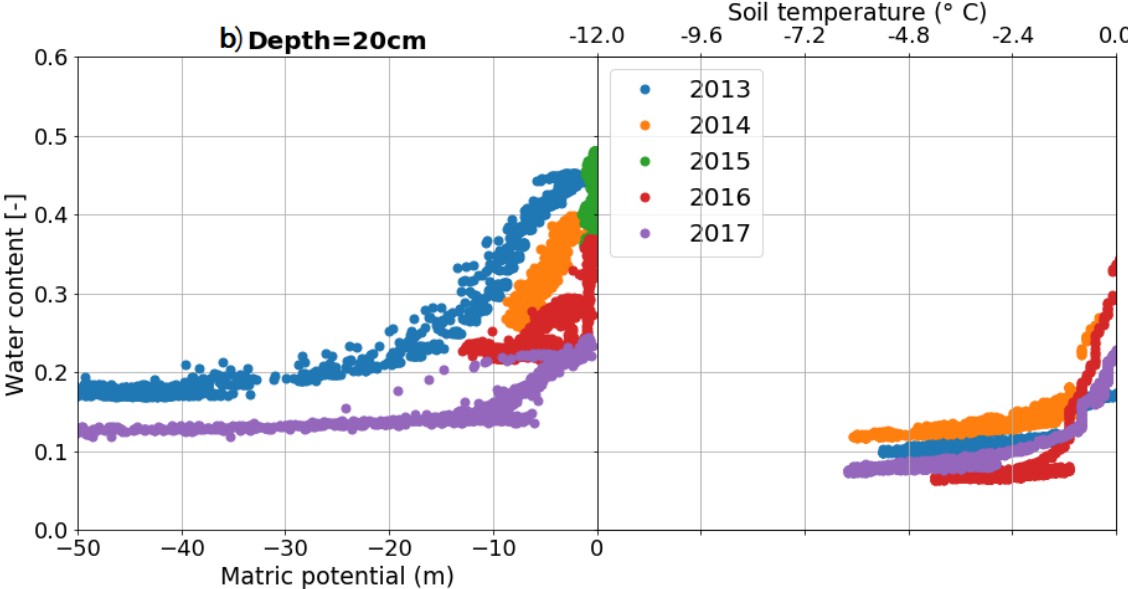

**Figure 7: Observed soil water content and soil freezing characteristic of the Upslope profile at 5 cm and 20 cm depths at St. Denis.**

Note that the water content data for the freezing condition of 2015 is not recorded in Fig. 7b since soil temperature at 20 cm depth was always above 0 $^0$c.

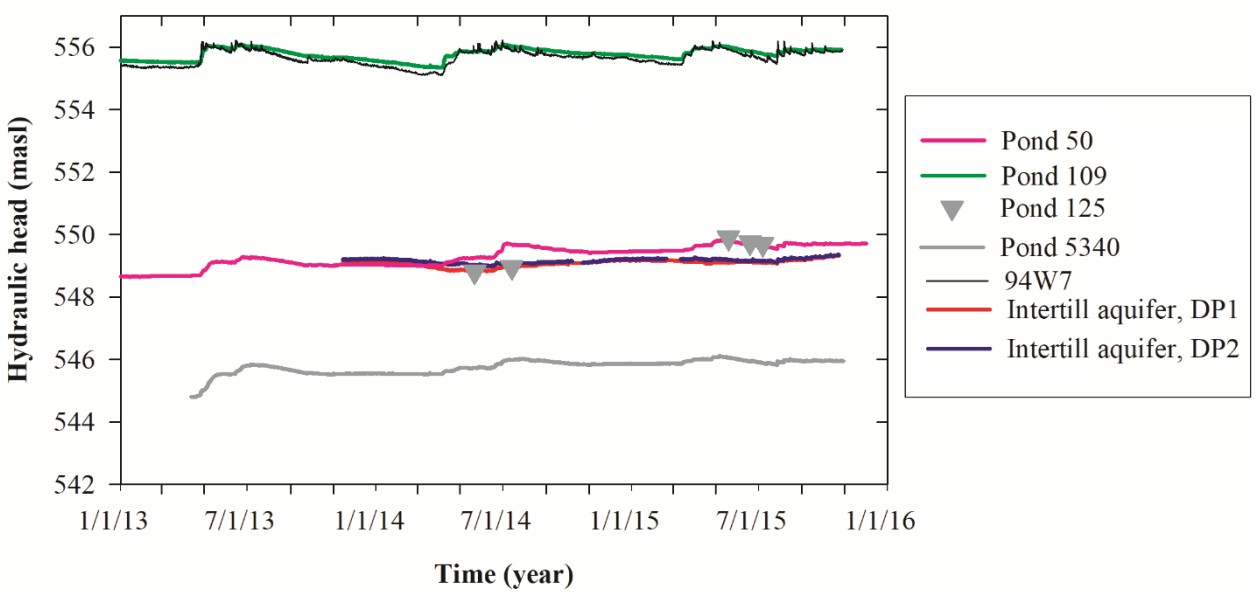

5 **Figure 8: Groundwater and surface water levels in St. Denis.**

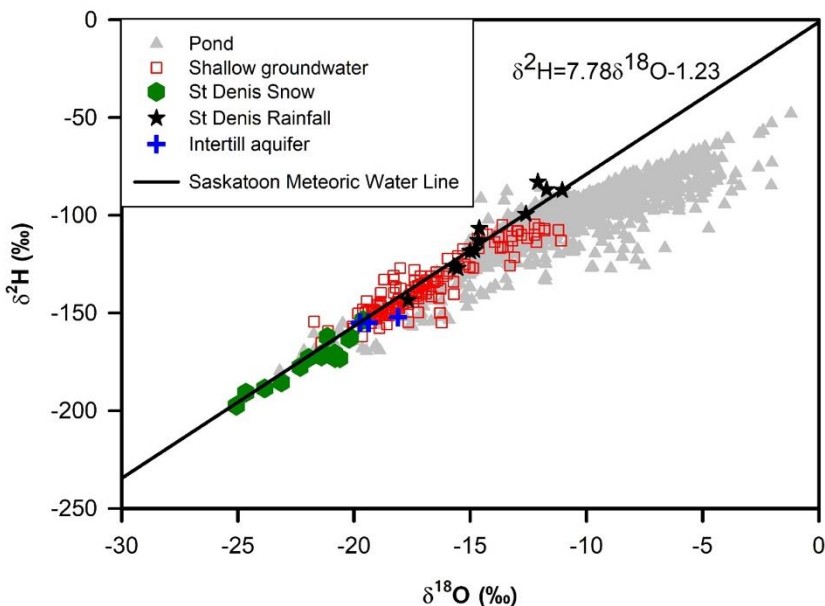

5    **Figure 9: A scatter plot of $\delta^2H$ versus $\delta^{18}O$ at St. Denis, Saskatchewan.**

**Tables**

5    **Table 1: Long-term meteorological instrumentation at St. Denis National Wildlife Area**

| 10 m mast | | |
| --- | --- | --- |
| *Variable* | *Height (m)* | *Sensor* |
| Wind speed (ms$^{-1}$) | 10 | RM Young 5103 |
| Wind direction (°) | 10 | RM Young 5103 |
| Air temperature/ Relative humidity (°C /%) | 2 | Vaisala HMP series |
| Photosynthetically active radiation (Wm$^{-2}$) | 1 | LiCor Li190SB |
| Incoming solar radiation (Wm$^{-2}$) | 1 | LiCor Li200X |
| Rainfall (mm) | 0.5 | Texas Electronics TE525M |
| | | |
| **10 m scaffolding tower** | | |
| *Variable* | *Height (m)* | *Sensor* |
| Wind speed (ms$^{-1}$) | 2 | Met One 14A |
| Wind direction (°) | 10 | NRG |
| Air temperature/ Relative humidity (°C /%) | 10 | Vaisala HMP series |
| Air temperature/ Relative humidity (°C /%) | 1.5 | Vaisala HMP series |
| All component radiation (Wm$^{-2}$) | 10 | Kipp and Zonen CNR4 |
| Turbulent fluxes (C, Qe, Qh) (Wm$^{-2}$) | 10 | Campbell Scientific CSAT3; EC150 |
| Rainfall (mm) | 10 | Texas Electronics TE525M |