# Peer review of "Meteorological, soil moisture, surface water, and groundwater data from the St. Denis National Wildlife Area, Saskatchewan, Canada"

_Earth System Science Data, 2018_

## Referee Comment (RC1) · Dr Nachshon (Referee) · 8 Nov 2018

This is a very useful document that describes the impressive monitoring work that has been done in the Canadian prairies, particularly at the St. Denis natural research area, in Saskatchewan.

Over the last decades, and mainly over the last few years a lot of environmental data has been collected at the St. Denis research site. The data includes meteorological data, hydrological information, soil measurements, isotopes analyses, and more. All of this data is well organized and available for any one who is willing to use it. The St. Denis site represents the natural (and cultivated) environment of the Canadian prairies

and the availability of the data to the scientific community is priceless.

The MS is well written and should be published.

I know that there is also chemical (major ions) information about the ponds water and I don't think the authors mentioned it. They briefly mentioned the stable isotopes but I would recommend the authors to elaborate on the chemistry information that they have.

On top of that - I found few minor typos:

P1, L20: "isotope isotopes" P3, L1: "biologist,s," P3, L2: "on," - delete the comma

At the isotopes section there are many acronyms and I'm not sure all of them were explained. Please check.

---

## Referee Comment (RC2) · W. Appels (Referee) · 19 Nov 2018

General comments

It is great to see the data of this prairie monitoring site being made available to all. As the authors mention, there are timeseries of varying observation frequency and duration. Together they paint a very detailed picture of hydrological dynamics of prairie wetlands and their surroundings. The dataset fits very well into the scope of EESD. Please consider the comments below to clarify some of the descriptions.

Specific comments

Page 3, line 5: Are you able to include the long-term pond chemistry data in the open dataset? If not, maybe mention that briefly here.

Figure 1: Inset with location of the site: what does the grey shading indicate?

Section 3: Please describe when in the winter the SWE data was collected. The wording of the caption of Figure 3 is not 100% clear to me: are the points the cumulative SWE of snow fallen or the content of the snow pack before melt? Are they averages of multiple days as well as of multiple locations in the landscape? If not, maybe include the date of each survey in the csv file as well.

Technical comments

page 3, line 1, typo: biologists

Figure 2 - temperature plots: use degree symbol in the axis labels

Figure 6: consider changing the point types/sizes to show some of the 2015 freezing characteristic. Or is it missing/compromised?

---

## Author Comment (AC1) · 20 Dec 2018

Reply to RC1

We thank the Referee #1, Dr. Nachshon for the encouraging comments on this manuscript.

We briefly mentioned this on page 3 line 5 as pointed out by the Referee #2- Dr. Appels, but the data was not included in the original submission. We have taken note of the importance, and usefulness of the data have thus elaborated on how this data is collected in the revised manuscript's Section 7 (page 8 Water isotope data and sur-

face water chemistry). We have also added a time-series of pond water chemistry plot Figure 5 and all the pond chemistry data in the dataset.

The concerns about (few minor typos: P1, L20: "isotope isotopes" P3, L1: "biologist,s," P3, L2: "on," - delete the comma At the isotopes section there are many acronyms and I'm not sure all of them were explained) have been addressed appropriately.

—————————————————————

[Figure]

**Fig. 1.** Figure 5

---

## Referee Comment (RC3) · Anonymous Referee #3 · 2 Jan 2019

General comments

The data set for St Denis NWA is very useful as it represents an unusually complete set of variables for the region. The atmospheric data are at high frequency, and include variables such as the turbulent fluxes, which are rarely found. The many researchers who laboured to collect the data are to be thanked for their hard work, as are the authors who have collected and presented the data sets.

Unfortunately, the writing is marked by vagueness and colloquialisms. The following terms need to be fixed:

By "average" you are usually referring to mean values.

[Figure]

Use of the words "high" and "low" is colloquial, unless you mean some type of elevation. You are referring to things which are either "large" or "small"

Comparative words like "colder", "higher", or "greater" imply that you are comparing a value to another value, which in many cases is not specified. This needs to be fixed.

Specific comments

Line 21

"ground elevations (datum) used". The term "elevations" is plural; "datum" is singular.

Line 29

"dominated by glacial till, as well as coarser grained fluvial deposits," This implies that the fluvial deposits are coarser than the till.

Line 2

"ponds that annually dry out" A better phrasing would be "ponds that dry out annually"

Line 3

"Farming practices over the past century include widespread artificial wetland drainage in some portions of the region." This statement should be backed up by a reference.

Line 6

"The site was selected because it was primarily a cultivated..."

A better phrasing would be

"The site was selected primarily because it was a cultivated..."

Line 12

"No wetland drainage has occurred on the site since 1968." Did any drainage occur before this year?

Line 27

"The area is hummocky" Which area? This is a poor word to use, as "area" has a mathematical meaning. Do you mean the general region, or the NWA?

Line 28

"for which detailed Lidar elevation data are available." Is this dataset available to other researchers? I don't see it in the provided data sets.

Line 1

"In the past 50 years" Would be better to replace "In" with "Over" as the activity continues to the present.

Line 11

As there are 2 stations, the word "oldest" should be replaced by "older"

Line 20

Insert the word "energy" after "turbulent" as the carbon fluxes are also turbulent. Insert the word "solar" after "net" to identify the type of radiation. Also, insert a dash between "all" and "component"

Line 24

The manufacturer of the logger is specified in Line 26, so it should be included here, too.

Line 26

According to Table 1, and to the headers of the .csv files, the "precipitation" data are

actually tipping bucket rainfalls – why would they be corrected for the effects of wind speed on snowfall? Please insert a complete description here of the data. Referring this data as "precipitation" is very confusing.

Line 7

Replace "second" with "other" before "snow survey", as the other snow survey site was not numbered.

Line 27

"Generally, monitoring typically" Pick one, either "generally" or "typically".

Line 4

"in (Conly et al., 2004)", should be "by Conly et al. (2004)"

Line 19

"For the wetlands" This is not required – delete.

Line 2

"During soil freezing the dielectric constant of ice is much lower than that of liquid water, so the instrument is likely to give a reasonable measure of the liquid water content"

This sentence is problematic. It is unclear whether you are referring to the liquid water phase in frozen soils or to the completely unfrozen soils. It needs to be re-written.

Replace "lower" with "smaller"

Insert "more" between "is" and "likely"

Line 3

What does the number in "229 probes" refer to? Is it the number of probes that were installed, or is it a model number?

Line 14

"shallow high permeability weathered till layers"

I think that it is important to indicate that the high permeability is relative to other types of till, so it would be a good idea to insert the word "relatively" before "high"

Line 27

"Stables" should be "Stable"

Line 4

"There is little lag in net radiation" What does this mean? What it is lagging with respect to? Are you saying that there is little seasonal lag between the incoming short-wave radiation and the net all-wave radiation?

Line 6

"Wind speeds average ..." Over what time periods? Are these daily values?

Line 20

"Wetlands 1, 50 and 109 are representative of prairie wetlands" What do the other ones represent, mountain wetlands?

Line 28

What is the "freeze back"? I am not familiar with this term.

Line 29

"The saturated water content at freeze-up is usually 0.5" What does this mean? What is the saturated water content 0.5 of? Do you mean that the saturation fraction is 0.5?

Figures

Figure 1

The soil moisture profile points near Pond 109 are so large that they overlap, so it is hard to see them all. It would be better to use smaller markers.

Figure 2

This figure is not referenced in the text.

Which set of air temperatures is plotted, the 2m or 5m? Please include the elevation, or the name of the site in the y-axis label. Also, include the time-step of the values plotted, i.e. either daily or monthly.

If you are going to refer to a value as a flux (the evaporation and precipitation) then it needs to have the units of a flux, i.e. as a mass (or depth) per unit time. It looks like you are plotting values which are accumulated over a time period.

The axis title in the bottom-right plot is confusing. As it states "Month", it implies that monthly values are plotted. It might be better to title it as simply "Date", as these are daily values. You could indicate the date format as "(Year-month)", if you like.

Figure 3

The blue line is the mean value. The use of the word "average" is confusing as a) it is incorrect and b) the individual SWE values are actually weighted averages.

Figure 4

"Hydrograph" should be plural.

Data sets

The headers of the isotope .csv files contain non-ASCII characters, which are problematic for many programs to read, particularly as there is no indication as to how the files are encoded. It would be a good idea to change these characters to their closest ASCII equivalents.

---

## Author Comment (AC2) · 11 Feb 2019

AUTHORS RESPONSE TO REVIEWER2

The authors reply (AC) to reviewer (RC) are numbered as AC1, AC2, AC3, . . . and RC1, RC2, RC3.... Reviewer:General comments RC1: It is great to see the data of this prairie monitoring site being made available to all. As the authors mention, there are timeseries of varying observation frequency and duration. Together they paint a very detailed picture of hydrological dynamics of prairie wetlands and their surroundings. The dataset fits very well into the scope of EESD. AC1:We thank the Referee # 2 Dr W. Appels for taking time to review the manuscript and also the encouraging comments

Please consider the comments below to clarify some of the descriptions. Specific comments RC2:Page 3, line 5: Are you able to include the long-term pond chemistry data in the open dataset? If not, may be mention that briefly here. AC2:We have included the long-term pond chemistry data and elaborated on the data collection and source in Section 7 of the revised manuscript. A new plot on the variation of pond EC among various ponds has also been added and brief note included under the data overview section (i.e, section 8). RC3: Figure 1: Inset with location of the site: what does the grey shading indicate? AC3: We thank the reviewer for this note. The grey shading is the prairie region of North America; this has been clarified in the Figure 1 label. RC4: Section 3: Please describe when in the winter the SWE data was collected. The wording of the caption of Figure 3 is not 100% clear to me: are the points the cumulative SWE of snow fallen or the content of the snow pack before melt? Are they averages of multiple days as well as of multiple locations in the landscape? If not, maybe include the date of each survey in the csv file as well. AC4: The snow survey is done once a year in the spring. We have added the date of the annual snow survey to the csv file. We have also updated the text in section 3 to indicate that the snow survey is performed once each year and that the values in figure 3 are an average SWE for the whole site. Figure 3 caption has been revised to indicate that the values shown are the average SWE of the snowpack before melt. RC5: Technical comments page 3, line 1, typo: biologists AC5: Typo corrected

RC6: Figure 2 - temperature plots: use degree symbol in the axis labels AC: Corrected RC7: Figure 6: consider changing the point types/sizes to show some of the 2015 freezing characteristic. Or is it missing/compromised? AC7: In 2015 there was negligible freezing at 20 cm, and only a small amount of freezing at 5 cm, as can be seen in Figure 5. Therefore Fig 6 provide an accurate depiction of the data in 2015. The data was not compromised or mising, rather it was not recorded, because the soil temperature at 20 cm below ground was always above 0 0c.

[Figure]

Figure 1: Map and location of St. Denis National Wildlife Area. The grey area on the inset map represents the extent of Prairie Pothole Region in North America.

**Fig. 1.** Figure 1: Map of study area

ESSDD

Interactive
comment

[Figure]

Figure 2: Meteorological data from SDNWA: a) the total annual rainfall for the period of record (1991-2017) and annual snowfall (only available 2014-2017); b) cumulative total precipitation (rain and snow) and evaporation for the 2016-2017 hydrologic year; c) mean monthly air temperature at 2 m for the period of record (1991-2017); and d) daily air temperature for the 2016-2017 hydrologic year.

**Fig. 2.** Figure 2: temperature -precipitation plots

---

## Author Comment (AC3) · 11 Feb 2019

AUTHORS RESPONSE TO REVIEWER 3

The authors reply (AC) to reviewer (RC)
high frequency, and include variables such as the turbulent fluxes, which are rarely found. The many researchers who laboured to collect the data are to be thanked for their hard work, as are the authors who have collected and presented the data sets. Unfortunately, the writing is marked by vagueness and colloquialisms. The following terms need to be fixed: By "average" you are usually referring to mean values. Use of the words "high" and "low" is colloquial, unless you mean some type of elevation. You are referring to things which are either "large" or "small" Comparative words like "colder", "higher", or "greater" imply that you are comparing a value to another value, which in many cases is not specified. This needs to be fixed. AC1: The authors are grateful for the comments by the reviewer and have made necessary changes to the language. Specific comments RC2: Page 1 Line 21 "ground elevations (datum) used". The term "elevations" is plural; "datum" is singular. AC2: This sentence has been edited for clarity. RC3: Line 29 "dominated by glacial till, as well as coarser grained fluvial deposits," This implies that the fluvial deposits are coarser than the till. RC3: This sentence has been edited to convey that there are sand/gravel deposits within the glacial till. RC4: Page 2 Line 2 "ponds that annually dry out" A better phrasing would be "ponds that dry out annually" Line 3 "Farming practices over the past century include widespread artificial wetland drainage in some portions of the region." This statement should be backed up by a reference. AC4: Reference added. RC5: Line 6 "The site was selected because it was primarily a cultivated..." A better phrasing would be "The site was selected primarily because it was a cultivated..." AC5: Agreed and fixed. RC6: Line 12 "No wetland drainage has occurred on the site since 1968." Did any drainage occur before this year? Line 27 "The area is hummocky" Which area? This is a poor word to use, as "area" has a mathematical meaning. Do you mean the general region, or the NWA? AC6: We mean both NWA and the region near the NWA. The sentence has been revised. RC7: Line 28 "for which detailed Lidar elevation data are available." Is this dataset available to other researchers? I don't see it in the provided data sets. AC7: We have added the LiDAR DEM data to the open dataset and included a short description of the dataset in a new section 8. RC8: Page 3 Line 1 "In the past 50 years"

Would be better to replace "In" with "Over" as the activity continues to the present. AC8: Accepted, replaced RC9: Line 11 As there are 2 stations, the word "oldest" should be replaced by "older" AC9: Ok, replaced RC10: Line 20 Insert the word "energy" after "turbulent" as the carbon fluxes are also turbulent. Insert the word "solar" after "net" to identify the type of radiation. Also, insert a dash between "all" and "component" AC10: Ok, replaced RC11: Line 24 The manufacturer of the logger is specified in Line 26, so it should be included here, too. AC: Ok, replaced. RC12: Line 26 According to Table 1, and to the headers of the .csv files, the "precipitation" data are actually tipping bucket rainfalls – why would they be corrected for the effects of wind speed on snowfall? Please insert a complete description here of the data. Referring this data as "precipitation" is very confusing. AC12: This paragraph has been changed to indicate that the precipitation data is measured with tipping bucket gauges, which only provide accurate rainfall data. This is why we call the data 'rainfall'. We have removed the word 'precipitation' from this description. However, the data we provide is raw and covers the entire year. Users will need to determine the precipitation phase using other data. RC13: Page 4 Line 7 Replace "second" with "other" before "snow survey", as the other snow survey site was not numbered. AC13: Ok, replaced RC14: Line 27 "Generally, monitoring typically" Pick one, either "generally" or "typically". AC14: agreed and change effected. RC15: Page 5 Line 4 "in (Conly et al., 2004)", should be "by Conly et al. (2004)" AC15: Ok. RC16: Line 19 "For the wetlands" This is not required – delete. AC16: Ok, deleted. RC17: Page 6 Line 2 "During soil freezing the dielectric constant of ice is much lower than that of liquid water, so the instrument is likely to give a reasonable measure of the liquid water content" This sentence is problematic. It is unclear whether you are referring to the liquid water phase in frozen soils or to the completely unfrozen soils. It needs to be re-written. Replace "lower" with "smaller" Insert "more" between "is" and "likely" AC17: The sentence was rewritten for clarity and refers to the liquid phase in frozen soil. The new sentence reads, "In frozen soil, the dielectric constant of ice is much smaller than that of liquid water, so the instrument is more likely to give a reasonable measure of the liquid water content".

RC18: Line 3 What does the number in "229 probes" refer to? Is it the number of probes that were installed, or is it a model number? AC18: The ''229 probes" here refer to the name of the heat dissipation sensors which measure the soil matric potential. The sensor is manufactured by the Campbell Scientific, Canada.

RC19: Line 14 "shallow high permeability weathered till layers" I think that it is important to indicate that the high permeability is relative to other types of till, so it would be a good idea to insert the word "relatively" before "high" AC19: Ok. RC20: Line 27 "Stables" should be "Stable" AC20: Ok. RC21: Page 8 Line 4 "There is little lag in net radiation" What does this mean? What it is lagging with respect to? Are you saying that there is little seasonal lag between the incoming short-wave radiation and the net all-wave radiation? AC21: Yes, that is what we are saying, and we have edited the sentence for clarity. RC22: Line 6 "Wind speeds average ..." Over what time periods? Are these daily values? AC22: This is the mean of hourly wind speed over the period of record. This has been clarified in the text. RC23: Line 20 "Wetlands 1, 50 and 109 are representative of prairie wetlands" What do the other ones represent, mountain wetlands? AC23: We have rephrased this sentence to avoid confusion. RC24: Line 28 What is the "freeze back"? I am not familiar with this term. AC24: The sentence was rephrased for clarity. The new sentence reads, "The frost table depth is also affected by antecedent soil water content".

RC25: Line 29 "The saturated water content at freeze-up is usually 0.5" What does this mean? What is the saturated water content 0.5 of? Do you mean that the saturation fraction is 0.5? AC25: The 'fraction' term was added in the sentence. The new sentence reads, "The saturated water content fraction at freeze –up is usually 0.5 with a high residual of liquid water content during frozen conditions".

Figures RC26: Figure 1 The soil moisture profile points near Pond 109 are so large that they overlap, so it is hard to see them all. It would be better to use smaller markers. AC26: We have added an inset detail map to the figure to show the soil moisture profiles near Pond 109. The profiles are too close together to show with non-overlapping

markers at the scale of the site map unless the markers are so small that they become difficult to see. RC27: Figure 2 This figure is not referenced in the text. Which set of air temperatures is plotted, the 2m or 5m? Please include the elevation, or the name of the site in the y-axis label. Also, include the time-step of the values plotted, i.e. either daily or monthly. If you are going to refer to a value as a flux (the evaporation and precipitation) then it needs to have the units of a flux, i.e. as a mass (or depth) per unit time. It looks like you are plotting values which are accumulated over a time. The axis title in the bottom-right plot is confusing. As it states "Month", it implies that monthly values are plotted. It might be better to title it as simply "Date", as these are daily values. You could indicate the date format as "(Year-month)", if you like. AC27: We have added a reference to this figure in the text in the Overview of content section. The figure has been revised showing temperature indicated in the plot was obtained at 2m.

RC28: Figure 3 The black line is the mean value. The use of the word "average" is confusing as a) it is incorrect and b) the individual SWE values are weighted averages. AC28: Ok. RC29: Figure 4 "Hydrograph" should be plural. AC29: Ok. Data sets RC 30: The headers of the isotope .csv files contain non-ASCII characters, which are problematic for many programs to read, particularly as there is no indication as to how the files are encoded. It would be a good idea to change these characters to their closest ASCII equivalents. AC30: Ok.
* * *
[Figure]

Figure 1: Map and location of St. Denis National Wildlife Area. The grey area on the inset map represents the extent of Prairie Pothole Region in North America.

**Fig. 1.** Figure 1: Map of study area

[Figure]

Figure 2: Meteorological data from SDNWA: a) the total annual rainfall for the period of record (1991-2017) and annual snowfall (only available 2014-2017); b) cumulative total precipitation (rain and snow) and evaporation for the 2016-2017 hydrologic year; c) mean monthly air temperature at 2 m for the period of record (1991-2017); and d) daily air temperature for the 2016-2017 hydrologic year.

**Fig. 2.** Figure 2: temperature -precipitation plots

---

## Author Response (AR3)

**Authors reply to Editor**

**Topical Editor Decision: Publish subject to minor revisions (review by editor)** (25 Feb 2019) by Chris DeBeer

Comments to the Author:

Dear authors,

Thank you for submitting your revised manuscript, for your efforts to address the reviewer comments, and for your additional contributions to the archived dataset. This is an improvement. However, there are a number of edits and corrections still required, as outlined below. I encourage the authors to carry out a careful review of the manuscript, as I had found many errors, but may have missed some.

Specific Comments:

| Editors comments | Authors response |
|---|---|
| Page 1, Abstract: Please remove the CCRN preamble from the abstract (lines 10-16). This is not relevant here and the site predates CCRN. | Done. |
| Page 1, Abstract: the abstract should be revised to better explain the context and history of the site and the data collection activities here, as well as describing the general characteristics of the dataset. | We have rewritten the abstract to address this point. |
| Page 1, line 17: What do you mean "for the prairie research site"? This needs to be revised. | NA – abstract revised. |
| Page 1, line 23: remove the "s" from "positions" | Done |
| Page 2, line 10: correct the phrase "…to ephemeral ponds that annually dry out annually." | Done |
| Page 2, line 21: the comment by reviewer #3 about wetland drainage has not been addressed. Was there any drainage before 1968? | We have changed the sentence to read "As far as we are aware, no wetland drainage has occurred on the site, though we cannot be certain about what happened prior to 1968 when the site was established as a National Research Area" |
| Page 2, line 29: the word "activities" should be added after "research" | Done. |
| Page 2, 3, Introduction: the introduction section jumps around a lot and needs to be better organized. i.e. paragraphs 2, 4, and 6 deal with the site and its history of research activity, while paragraphs 1,3, and 5 deal with the landscape and physiographic setting. | We have tried to improve the structure of the introduction. We start by introducing the prairies (para 1), then we introduce the site (para 2) then we talk about land use (para 3) and soils/geology (para |

| | |
|---|---|
| | 4), then we end with a summary research at the site and the focus of the paper (para 5). |
| Page 3, line 28: be careful about the term "net solar radiation". Is this correct? I'm not sure, as the CNR4 measures the balance of incoming and outgoing short- and long-wave radiation. | The editor is correct - the CNR4 measures both longwave and shortwave components of radiation. We have deleted "solar". |
| Page 4, line 9 and Figure 2: Where does the snowfall data come from? There is mention later on of a Geonor installed in 2015, but where is the 2014-17 snowfall data from? | The Geonor was actually installed in 2014, which we have corrected in the text. We have also removed the snowfall data from the plot because it is not part of the published dataset. |
| Page 4, line 18: Could you add a reference for the ESC snow tube? See https://agupubs.onlinelibrary.wiley.com/doi/epdf/10.1002/2015RG000481 for more information. | Done. |
| Page 5, line 26 and Figure 4: I am not sure it is appropriate to call these hydrographs as the graphs show pond level fluctuations over time. A hydrograph shows discharge rate against time. | Agreed. We changed the caption to "Pond level fluctuations over time in Wetlands 1, 50 and 109" |
| Page 6, line 22: the "229 probes" reference could be clearer, following a comment by reviewer #3. | A reference to the instruction manual was added. |
| Page 6, line 26: Fig 5 shows time series of EC. This should refer to Fig. 6. | Changed and checked all other references to figures. |
| Page 6, line 27: Where is transect 1 in Fig 1? And what is this, the snow survey transect? This should be more clear. | This should actually refer to soil profile 1. We have added numbers to the soil profile markers in Figure 1. |
| Page 7, lines3-4: the phrase "…comprising a shallow and relatively high permeability weathered till layers…" does not make sense. Is this plural? Then remove "a". | Removed "a" to correct this. |
| Page 7, line 11: Delete "The" before "Figure 8" | Done. |
| Page 8, paragraph 2: What is the difference between V-SMOW and V-SMOW2, and SLAP and SLAP2? This could be more clear. | These are different reference waters, and we distinguish them for precision. This is clear in |

| | |
|---|---|
| | the references we provide. See note * below for further details. |
| Page 9, line 10-11: Why is there a reference to Fig. 9? Should this be to Fig. 2 instead? | Corrected to Figure 2 |
| Page 9, line 23: again, I question whether these should be called hydrographs. | Corrected |
| Page 10, line 2: why is there a reference to Fig. 4, which shows pond level fluctuations? | This should have referred to Figure 5. It has been changed (and all other references checked for accuracy). |
| Page 10, line 10: the comment by reviewer #3 was not fully addressed. The correction was to have included the word "fraction" after "saturated water content" | The error was the water "saturated" – i.e. we are referring to the water content at freeze-up. This has been corrected. |
| Page 11: Acknowledgment: Please acknowledge the efforts of the Canadian Consortium for Lidar Environmental Applications Research (C-CLEAR) team for the lidar acquisition. | Done. |
| Page 17, Figure 2: you should be more clear what station the data plotted in each panel are from. Is the data in part (d) from a 2 m height? | Air temperature in parts (c) and (d) was measured at 2 m at the mast tower. The rainfall data is from the mast tower and evaporation data from the scaffold tower. The caption has been amended. |
| **Dataset:** | |
| Reviewer #3 had a comment about the headers of the .csv files having non-ASCII characters. Please clarify what was done in response and what changes were made. | The isotope files initially included a per mil symbol, which has been removed and replaced with the word "permil". |

* Regarding the difference between V-SMOW and V-SMOW2 and SLAP and SLAP2, the Consultants' Meeting on Stable Isotope Standards and Intercalibration in Hydrology and in Geochemistry recommended that oxygen and hydrogen isotope ratios be normalized on the VSMOW-SLAP scale, where the consensus value for $\delta^{18}O$ and $\delta^{2}H$ SLAP (V-SMOW: $\delta^{2}H$ = 0.0 and $\delta^{18}O$ = 0.0) and Standard Light Antarctic Precipitation (SLAP: $\delta^{2}H$ = –428.0 ‰ and $\delta^{18}O$ = –55.5 ‰). Due to the consumption of the original VSMOW and SLAP reference waters, two replacement reference waters, VSMOW2 and SLAP2, were developed. No significant difference between the original and replacement water $\delta^{18}O$ and $\delta^{2}H$ values has been detected and therefore VSMOW and SLAP are considered synonymous with VSMOW2 and SLAP2 when referring to the normalization of waters on the VSMOW-SLAP scale.

[revised manuscript text omitted]

**Figures**
20 **Figures**
**Figures**
**Figures**
**Figures**
**Figures**
25 **Figures**
**Figures**
**Figures**
**Figures**
**Figures**
30 **Figures**
**Figures**
**Figures**
**Figures**
**Figures**

[Figure]

**Figure 1: Map and location of St. Denis National Wildlife Area. The grey area on the inset map represents the extent of Prairie Pothole Region in North America**

[Figure]

**Figure 2: Meteorological data from SDNWA: a) the total annual rainfall for the period of record (1991-2017) measured at the mast tower; b) cumulative total rainfall (mast tower) and evaporation (scaffold tower) for the 2016-2017 hydrologic year; c) mean monthly air temperature at 2 m (mast tower) for the period of record (1991-2017); and d) daily air temperature at 2 m (mast tower) for the 2016-2017 hydrologic year.**

[Figure]

**Figure 3: Weighted mean snow water equivalent (SWE, blue bars) of snowpack before melt over SDNWA. The black line represents the 24-year average of snow accumulation.**

[Figure]

**Figure 4: Pond level fluctuations with time in Wetlands 1, 50 and 109.**

[Figure]

Figure 5: Time series of pond water electrical conductivity (log scale) at St. Denis showing the variations in pond salinity.

[Figure]

**Figure 6: Soil freezing depths in three soil profiles near Pond 109.**

[Figure]

[Figure]

5 **Figure 7: Observed soil water content and soil freezing characteristic of the Upslope profile at 5 cm and 20 cm depths at St. Denis.**

Note that the water content data for the freezing condition of 2015 is not recorded in Fig. 7b since soil temperature at 20 cm depth was always above 0 $^0$c.

[Figure]

5    **Figure 8: Groundwater and surface water levels in St. Denis.**

[Figure]

5    **Figure 9: A scatter plot of δ²H versus δ¹⁸O at St. Denis, Saskatchewan.**

**Tables**

Table 1: Long-term meteorological instrumentation at St. Denis National Wildlife Area

| 10 m mast | | |
|---|---|---|
| *Variable* | *Height (m)* | *Sensor* |
| Wind speed (ms$^{-1}$) | 10 | RM Young 5103 |
| Wind direction (°) | 10 | RM Young 5103 |
| Air temperature/ Relative humidity (°C /%) | 2 | Vaisala HMP series |
| Photosynthetically active radiation (Wm$^{-2}$) | 1 | LiCor Li190SB |
| Incoming solar radiation (Wm$^{-2}$) | 1 | LiCor Li200X |
| Rainfall (mm) | 0.5 | Texas Electronics TE525M |
| | | |
| **10 m scaffolding tower** | | |
| *Variable* | *Height (m)* | *Sensor* |
| Wind speed (ms$^{-1}$) | 2 | Met One 14A |
| Wind direction (°) | 10 | NRG |
| Air temperature/ Relative humidity (°C /%) | 10 | Vaisala HMP series |
| Air temperature/ Relative humidity (°C /%) | 1.5 | Vaisala HMP series |
| All component radiation (Wm$^{-2}$) | 10 | Kipp and Zonen CNR4 |
| Turbulent fluxes (C, Qe, Qh) (Wm$^{-2}$) | 10 | Campbell Scientific CSAT3; EC150 |
| Rainfall (mm) | 10 | Texas Electronics TE525M |

**AUTHORS RESPONSE TO REVIEWERS**

**The authors reply to reviewers are in bullet and highlighted in red**

**REVIEWER 1:**

This is a very useful document that describes the impressive monitoring work that has been done in the Canadian prairies, particularly at the St. Denis natural research area, in Saskatchewan. Over the last decades, and mainly over the last few years a lot of environmental data has been collected at the St. Denis research site. The data includes meteorological data, hydrological information, soil measurements, isotopes analyses, and more. All of this data is well organized and available for any one who is willing to use it. The St. Denis site represents the natural (and cultivated) environment of the Canadian prairies and the availability of the data to the scientific community is priceless. The MS is well written and should be published.

- We thank the Referee #1, Dr. Nachshon for the encouraging comments on this manuscript.

I know that there is also chemical (major ions) information about the ponds water and I do not think the authors mentioned it. They briefly mentioned the stable isotopes, but I would recommend the authors to elaborate on the chemistry information that they have.

This is a good suggestion, and we have now included pond water chemistry data in the revised data set. We have taken note of the importance and usefulness of the data and have thus elaborated on how this data is collected in the revised manuscript's Section 7 (Water isotope data and surface water chemistry) page 8.

- On top of that - I found few minor typos: P1, L20: "isotope isotopes" P3, L1: "biologist,s," P3, L2: "on," - delete the comma At the isotopes section there are many acronyms and I'm not sure all of them were explained.
- These corrections have been made.

**REVIEWER 2:**

General comments

It is great to see the data of this prairie monitoring site being made available to all. As the authors mention, there are timeseries of varying observation frequency and duration. Together they paint a very detailed picture of hydrological dynamics of prairie wetlands and their surroundings. The dataset fits very well into the scope of EESD.

- We thank the Referee # 2 Dr W. Appels for taking time to review the manuscript and also the encouraging comments

Please consider the comments below to clarify some of the descriptions. Specific comments

Page 3, line 5: Are you able to include the long-term pond chemistry data in the open dataset? If not, may be mention that briefly here.

- We have included the long-term pond chemistry data and elaborated on the data collection and source in Section 7 of the revised manuscript. A new plot on the variation of pond EC among various ponds has also been added and brief note included under the data overview section (i.e, section 8).

Figure 1: Inset with location of the site: what does the grey shading indicate?

- We thank the reviewer for this note. The grey shading is the prairie region of North America; this has been clarified in the Figure 1 label.

Section 3: Please describe when in the winter the SWE data was collected. The wording of the caption of Figure 3 is not 100% clear to me: are the points the cumulative SWE of snow fallen or the content of the snow pack before melt? Are they averages of multiple days as well as of multiple locations in the landscape? If not, maybe include the date of each survey in the csv file as well.

- The snow survey is done once a year in the spring. We have added the date of the annual snow survey to the csv file. We have also updated the text in section 3 to indicate that the snow survey is performed once each year and that the values in figure 3 are an average SWE for the whole site.
- Figure 3 caption has been revised to indicate that the values shown are the average SWE of the snowpack before melt.

Technical comments page 3, line 1, typo: biologists

- Typo corrected

Figure 2 - temperature plots: use degree symbol in the axis labels

- Corrected

Figure 6: consider changing the point types/sizes to show some of the 2015 freezing characteristic. Or is it missing/compromised?

- In 2015 there was negligible freezing at 20 cm, and only a small amount of freezing at 5 cm, as can be seen in Figure 5. Therefore Fig 6 provide an accurate depiction of the data in 2015.
- The data was not compromised or miising, rather it was not recorded, because the soil temperature at 20 cm below ground was always above 0 $^0$c.

**REVIEWER 3:**
The data set for St Denis NWA is very useful as it represents an unusually complete set of variables for the region. The atmospheric data are at high frequency, and include variables such as the turbulent fluxes, which are rarely found. The many researchers who laboured to collect the data are to be thanked for their hard work, as are the authors who have collected and presented the data sets. Unfortunately, the writing is marked by vagueness and colloquialisms.

The following terms need to be fixed: By "average" you are usually referring to mean values.

Use of the words "high" and "low" is colloquial, unless you mean some type of elevation. You are referring to things which are either "large" or "small" Comparative words like "colder", "higher", or "greater" imply that you are comparing a value to another value, which in many cases is not specified. This needs to be fixed.

- The authors are grateful for the comments by the reviewer and have made necessary changes to the language.

Specific comments

**Page 1** Line 21 "ground elevations (datum) used". The term "elevations" is plural; "datum" is singular.

- This sentence has been edited for clarity.

Line 29 "dominated by glacial till, as well as coarser grained fluvial deposits," This implies that the fluvial deposits are coarser than the till.

- This sentence has been edited to convey that there are sand/gravel deposits within the glacial till.

**Page 2** Line 2 "ponds that annually dry out" A better phrasing would be "ponds that dry out annually" Line 3 "Farming practices over the past century include widespread artificial wetland drainage in some portions of the region." This statement should be backed up by a reference.

- Reference added.

Line 6 "The site was selected because it was primarily a cultivated..." A better phrasing would be "The site was selected primarily because it was a cultivated..."

- Agreed and fixed.

Line 12 "No wetland drainage has occurred on the site since 1968." Did any drainage occur before this year? Line 27 "The area is hummocky" Which area? This is a poor word to use, as "area" has a mathematical meaning. Do you mean the general region, or the NWA?

- We mean both NWA and the region near the NWA. The sentence has been revised.

Line 28 "for which detailed Lidar elevation data are available." Is this dataset available to other researchers? I don't see it in the provided data sets.

- We have added the LiDAR DEM data to the open dataset and included a short description of the dataset in a new section 8.

**Page 3** Line 1 "In the past 50 years" Would be better to replace "In" with "Over" as the activity continues to the present.

- Accepted, replaced

Line 11 As there are 2 stations, the word "oldest" should be replaced by "older"

- Ok, replaced

Line 20 Insert the word "energy" after "turbulent" as the carbon fluxes are also turbulent. Insert the word "solar" after "net" to identify the type of radiation. Also, insert a dash between "all" and "component"

- Ok, replaced

Line 24 The manufacturer of the logger is specified in Line 26, so it should be included here, too.

- Ok, replaced.

Line 26 According to Table 1, and to the headers of the .csv files, the "precipitation" data are actually tipping bucket rainfalls – why would they be corrected for the effects of wind speed on snowfall? Please insert a complete description here of the data. Referring this data as "precipitation" is very confusing.

- This paragraph has been changed to indicate that the precipitation data is measured with tipping bucket gauges, which only provide accurate rainfall data. This is why we call the data 'rainfall'. We have removed the word 'precipitation' from this description. However, the data we provide is raw and covers the entire year. Users will need to determine the precipitation phase using other data.

**Page 4** Line 7 Replace "second" with "other" before "snow survey", as the other snow survey site was not numbered.

- Ok, replaced

Line 27 "Generally, monitoring typically" Pick one, either "generally" or "typically".

- agreed and change effected.

Page 5 Line 4 "in (Conly et al., 2004)", should be "by Conly et al. (2004)"

- Ok.

Line 19 "For the wetlands" This is not required – delete.

- Ok, deleted.

**Page 6** Line 2 "During soil freezing the dielectric constant of ice is much lower than that of liquid water, so the instrument is likely to give a reasonable measure of the liquid water content" This sentence is problematic. It is unclear whether you are referring to the liquid water phase in frozen soils or to the completely unfrozen soils. It needs to be re-written. Replace "lower" with "smaller" Insert "more" between "is" and "likely"

- The sentence was rewritten for clarity and refers to the liquid phase in frozen soil. The new sentence reads, "In frozen soil, the dielectric constant of ice is much smaller than that of liquid water, so the instrument is more likely to give a reasonable measure of the liquid water content".

Line 3 What does the number in "229 probes" refer to? Is it the number of probes that were installed, or is it a model number?

- The ''229 probes'' here refer to the name of the heat dissipation sensors which measure the soil matric potential. The sensor is manufactured by the Campbell Scientific, Canada.

Line 14 "shallow high permeability weathered till layers" I think that it is important to indicate that the high permeability is relative to other types of till, so it would be a good idea to insert the word "relatively" before "high"

- Ok.

Line 27 "Stables" should be "Stable"

- Ok.

**Page 8** Line 4 "There is little lag in net radiation" What does this mean? What it is lagging with respect to? Are you saying that there is little seasonal lag between the incoming short-wave radiation and the net all-wave radiation?

- Yes, that is what we are saying, and we have edited the sentence for clarity.

Line 6 "Wind speeds average ..." Over what time periods? Are these daily values?

- This is the mean of hourly wind speed over the period of record. This has been clarified in the text.

Line 20 "Wetlands 1, 50 and 109 are representative of prairie wetlands" What do the other ones represent, mountain wetlands?

- We have rephrased this sentence to avoid confusion.

Line 28 What is the "freeze back"? I am not familiar with this term.

- The sentence was rephrased for clarity. The new sentence reads, "The frost table depth is also affected by antecedent soil water content".

Line 29 "The saturated water content at freeze-up is usually 0.5" What does this mean? What is the saturated water content 0.5 of? Do you mean that the saturation fraction is 0.5?

- The 'fraction' term was added in the sentence. The new sentence reads, "The saturated water content fraction at freeze –up is usually 0.5 with a high residual of liquid water content during frozen conditions".

**Figures**

Figure 1 The soil moisture profile points near Pond 109 are so large that they overlap, so it is hard to see them all. It would be better to use smaller markers.

- We have added an inset detail map to the figure to show the soil moisture profiles near Pond 109. The profiles are too close together to show with non-overlapping markers at the scale of the site map unless the markers are so small that they become difficult to see.

Figure 2 This figure is not referenced in the text. Which set of air temperatures is plotted, the 2m or 5m? Please include the elevation, or the name of the site in the y-axis label. Also, include the time-step of the values plotted, i.e. either daily or monthly. If you are going to refer to a value as a flux (the evaporation and precipitation) then it needs to have the units of a flux, i.e. as a mass (or depth) per unit time. It looks like you are plotting values which are accumulated over a time. The axis title in the bottom-right plot is confusing. As it states "Month", it implies that monthly values are plotted. It might be better to title it as simply "Date", as these are daily values. You could indicate the date format as "(Year-month)", if you like.

- We have added a reference to this figure in the text in the Overview of content section. The figure has been revised….

Figure 3 The blue line is the mean value. The use of the word "average" is confusing as a) it is incorrect and b) the individual SWE values are weighted averages.

- Ok.

Figure 4 "Hydrograph" should be plural.

- Ok.

**Data sets**

The headers of the isotope .csv files contain non-ASCII characters, which are problematic for many programs to read, particularly as there is no indication as to how the files are encoded. It would be a good idea to change these characters to their closest ASCII equivalents.

- Ok.

[revised manuscript text omitted]